# Serum Biomarker Analysis in Pediatric ADHD: Implications of Homocysteine, Vitamin B12, Vitamin D, Ferritin, and Iron Levels

**DOI:** 10.3390/children11040497

**Published:** 2024-04-22

**Authors:** Tanja Lukovac, Olivera Aleksić Hil, Milka Popović, Vitomir Jovanović, Tatjana Savić, Aleksandra M. Pavlović, Dragan Pavlović

**Affiliations:** 1Center for Speech and Language Pathology Higia Logos, Mirijevski Bulevar 17 b, 11060 Belgrade, Serbia; 2Institute of Mental Health, Palmoticeva 37, 11000 Belgrade, Serbia; oalexic@gmail.com; 3Beo-Lab Laboratories, Resavska 58-60, 11000 Belgrade, Serbia; milka.popovic@beo-lab.rs; 4Faculty of Philosophy, University of Belgrade, Čika-Ljubina 18-20, 11102 Belgrade, Serbia; vitomirj@gmail.com; 5Institute for Biological Research “Siniša Stanković”—National Institute of the Republic of Serbia, University of Belgrade, 142 Despot Stefan Boulevard, 11000 Belgrade, Serbia; tanjat@ibiss.bg.ac.rs; 6Faculty of Special Education and Rehabilitation, University of Belgrade, Visokog Stevana 2, 11102 Belgrade, Serbia; aleksandra3003@yahoo.com (A.M.P.); dpavlovic53@hotmail.com (D.P.)

**Keywords:** ADHD, homocysteine, iron, ferritin, vitamin B12, vitamin D, children

## Abstract

The current diagnosis of attention deficit hyperactivity disorder (ADHD) is based on history, clinical observation, and behavioral tests. There is a high demand to find biomarkers for the diagnosis of ADHD. The aim of this study is to analyze the serum profiles of several biomarkers, including homocysteine (Hcy), vitamin B12, vitamin D, ferritin, and iron, in a cohort of 133 male subjects (6.5–12.5 years), including 67 individuals with an ADHD diagnosis based on DSM-V criteria and 66 age-matched healthy boys (healthy controls, HC). Assessments for ADHD included the Iowa Conners’ Teacher Rating Scale (CPRS) and the ADHDT test, as well as cognitive assessments using the Wechsler Intelligence Scale for Children-Revised (WISC-R) and the TROG-2 language comprehension test. Hcy and iron were quantified using spectrophotometry, while vitamin B12 and total 25-hydroxy vitamin D levels were determined using an electrochemiluminescence immunoassay (ECLIA) and ferritin was measured using a particle-enhanced immunoturbidimetric assay. The results showed significantly increased Hcy levels and decreased vitamin B12 levels in ADHD patients compared to HCs. Multiple logistic regression analysis indicated that Hcy is a potential prognostic indicator for ADHD. These results suggest that elevated homocysteine and decreased vitamin B12 may serve as markers for the diagnosis and prognosis of ADHD.

## 1. Introduction

Attention deficit hyperactivity disorder (ADHD) is a neurodevelopmental disorder defined by impaired levels of inattention, disorganization, and/or hyperactivity/impulsivity [1]. With a prevalence of 7.2% in children and adolescents aged 6–17 years and 3–5% in adults [2,3], ADHD is one of the most common neurobehavioral disorders. ADHD is more common in boys than in girls [4]. In addition to the most common symptoms such as inattention, impulsivity, impaired inhibition, and hyperactivity, patients with ADHD may also show deficits in higher cognitive functions [5]. People with ADHD are at increased risk for other psychiatric disorders and often experience academic and occupational failure, criminal behavior, social disability, and addiction issues [6]. It has been found that some ADHD symptoms may correlate with certain hormonal patterns such as TSH level and symptoms of hyperactivity and impulsivity [7]. The pathophysiological mechanisms of ADHD are still not understood. Genetic, psychological, and environmental factors are generally accepted as causes of this disorder and could explain the etiology of ADHD [8,9].

The diagnosis of ADHD involves several crucial steps that require the collection of information from various sources in order to comprehensively assess a child’s behavior. This involves the parents, the child, school teachers, and other family members. The Conners Parent Rating Scale [10] usually serves as the cornerstone for identifying ADHD and assessing the severity of symptoms. In addition, the ADHD Test (ADHDT), developed by Gilliam in 1995, is another valuable tool for identifying individuals who exhibit ADHD characteristics. These diagnostic measures enable a holistic understanding of the manifestations of the disorder and help to formulate customized intervention strategies. The treatment of ADHD includes medication and/or behavioral therapy. Starting treatment early increases its effectiveness and improves the prognosis [3,11]. Psychostimulants can help, but are associated with side effects such as loss of appetite, headaches, stomach pain, agitation, sleep disturbances, anxiety, and insomnia [12,13]. Methylphenidate, a standard treatment for ADHD, is associated with many side effects, including insomnia and decreased appetite, especially in the first six months of use; this often leads to the discontinuation of the medication [14,15]. A growing number of studies suggest that neuroinflammation and oxidative stress play a crucial role in the pathophysiology of ADHD. These studies have focused on natural products as novel, promising ADHD treatments with fewer side effects [12,16]. The spectrum of potential factors influencing ADHD symptoms includes metabolic abnormalities and nutritional deficiencies [17,18]. As a result, there is growing interest in researching alternative therapies for ADHD, with a focus on the neuroprotective properties of natural products and substances. These alternatives are gaining traction due to their acceptance by patients and their perceived health benefits over conventional pharmaceutical interventions [19].

Homocysteine (Hcy) is an important thiol-containing amino acid that is formed by the intracellular demethylation of methionine. It is part of the methionine cycle, which is closely linked to S-adenosylmethionine, vitamin B12, and folic acid. Elevated Hcy levels in the blood often indicate a disruption of the methylation pathways, which can have various medical consequences [20,21]. Extensive research has emphasized the link between elevated Hcy levels and a spectrum of neurodegenerative diseases, cognitive impairment, and psychiatric conditions [22,23,24,25,26,27]. In ADHD in particular, Hcy is implicated in the induction of oxidative stress, the modulation of oxygen levels, and the initiation of lipid peroxidation pathways [28]. It is also assumed that a disturbed Hcy metabolism impairs cognitive functions [29]. The concentration of Hcy in the blood is closely linked to the levels of various B vitamins, in particular, cyanocobalamin (B12), folic acid (B9), and pyridoxine (B6) [30]. There is a strong correlation between high Hcy levels and vitamin B12 or folate [31], which usually indicates a deficiency of vitamin B12 or folate [30]. Such deficits are associated with the etiology of neurodevelopmental disorders such as ADHD and autism [32,33,34,35]. Folate in particular is essential for the normal development of the nervous system, as it plays a regulatory role in neurogenesis [36]. In addition, both genetic and clinical evidence suggest that folate and Hcy are involved in the development of psychiatric disorders [25].

Recently, vitamin D deficiency has been associated with psychiatric disorders, such as autism, depression, schizophrenia, and ADHD [33,37,38,39,40,41,42]. Eyles et al. found that vitamin D can have neurohormonal properties in the human brain [43]. A study by Neyestani et al. found that vitamin D deficiency is prevalent in Iran, especially in school-age children [44]. The early detection of vitamin D deficiency is essential in children with ADHD. Vitamin D can be used as an adjunct therapy to methylphenidate to reduce ADHD symptoms [11,45].

Iron, which is considered a crucial cofactor for enzymes involved in the pathophysiology of ADHD [41], presumably plays a central role in dopamine synthesis and could, therefore, influence ADHD symptoms [46]. A deficiency of this mineral has been associated with cognitive and behavioral disorders in children, particularly in the form of attention deficits and hyperactivity [47]. Despite numerous studies, the relationship between serum ferritin levels and ADHD remains unclear, with conflicting results regarding the correlation between serum ferritin and brain iron levels [48]. The intricate interplay between nutritional factors and genetic pathways critical to brain development, social cognition, and decision-making remains poorly understood, as does the potential for these gene–environment interactions to trigger mental disorders [49]. Consequently, there is an urgent need for further research to clarify the status of key biochemical parameters, including Hcy, vitamin B12, vitamin D, iron, and ferritin, in children with ADHD. It is noteworthy that, to our knowledge, no study in Serbia has yet investigated the levels of these biochemical parameters in children diagnosed with ADHD. This study aims to fill this gap by investigating the levels of Hcy, vitamin B12, vitamin D, iron, and ferritin in 6.5–12.5-year-old male subjects diagnosed with ADHD and exploring their correlation with symptom severity.

## 2. Materials and Methods

The study presented here is a causal study in which a connection was sought between certain serum biomarkers and the severity of ADHD symptoms. As part of a cohort study design, the study participants were divided into ADHD and control groups according to age, serum samples were taken for biochemical analyses, and the data were statistically analyzed as described below.

This study comprised 133 male participants aged between 6.5 and 12.5 years, including 67 individuals diagnosed with ADHD from the Institute of Mental Health in Belgrade and 66 healthy controls (HC) from the general population. The healthy subjects were carefully matched for age according to the recommendations of pediatric experts. To ensure that no neurodevelopmental problems, including ADHD, were present, family history examinations were performed on the HC participants. Exclusion criteria included bilingualism; obvious somatic, neurological, psychiatric, or sensory disorders; and a below average intellectual level. Participation in the study was strictly voluntary and all subjects were enrolled after receiving written parental consent.

The parents of the subjects diagnosed with ADHD completed the revised long form of the Conners Parental Rating Scale, while their respective teachers used the Conners Teacher Rating Scale for assessment. The validity of these scales has already been established by Dereboy [50]. In addition, ADHD symptoms were assessed by neuropsychiatrists using Gilliam’s ADHDT tests (1995). All participants underwent comprehensive cognitive testing, including the Wechsler Intelligence Scale for Children-Revised (WISC-R) [51], to ensure that individuals with below average intellectual ability were excluded. Participants in both groups had an intelligence quotient (IQ) of ≥90 according to the WISC-R criteria. In addition, the Test for the Reception of Grammar, Version 2 (TROG-2) [52], was administered to assess participants’ comprehension of grammatical categories to confirm the presence or absence of receptive language deficits. This language assessment was critical to ensure that all participants consistently understood the verbal instructions. Both the WISC-R and TROG-2 tests were administered 3–5 days prior to blood collection. Subjects were not taking pharmacotherapeutic medications or supplements for at least two months prior to testing and blood sampling. This study lasted eight months, from October to May, with the summer months deliberately avoided in order to minimize the influence of seasonal fluctuations in sunlight on vitamin D uptake [53].

Ethical approval for this study was obtained from the Ethics Committee of the Institute of Mental Health in Belgrade (approval number, 1704/1). Written informed consent was obtained from the parents of all children who participated in this study. All participants gave their consent 3 to 5 days before the test.

The experimental tests were carried out in a quiet, well-lit room. Pens, an evaluation sheet, and other aids were provided. To prevent fatigue, the examination was conducted in a single session with breaks. Testing began with a short interview in which the children’s daily routine, diet, and nutritional supplements were enquired about, and compliance with inclusion and exclusion criteria was determined. The participants were well prepared for the blood sampling, so that the fear of blood sampling was minimized. A venous blood sample was taken at 9 am; the subjects had not had any food or drink the previous night or morning. The necessary laboratory materials for blood sampling and the storage of the blood were provided.

Hcy, vitamin B12, vitamin D, iron, and ferritin were measured in blood serum. Blood samples (minimum volume of 4 mL) were taken by venipuncture and collected in a biochemical test tube with gel. After blood collection, the test tube was set aside for 15 min to allow clotting, before centrifugation at 1800× *g* for 10 min. After separation from the serum, the samples were stored at −20 °C and then analyzed. The total Hcy in human serum was quantified using an Hcy Enzymatic Assay Kit on a Roche C311 biochemical instrument. Enzyme cycling was measured spectrophotometrically at 340 nm. The Hcy concentration in the sample is directly proportional to the amount of NADH (nicotinamide adenine dinucleotide + hydrogen), which is converted to NAD+ (nicotinamide adenine dinucleotide). Vitamin B12 and the total 25-hydroxy vitamin D in serum were measured by an electrochemiluminescence immunoassay (ECLIA) on Elecsys and Cobas^®^ immunoassay analyzers. The results were determined using a calibration curve generated specifically for each analyzer by a two-point calibration and a master curve provided by the reagent barcode. Iron and ferritin in human serum were quantified in vitro on Roche/Hitachi Cobas C311 systems. Iron was determined using a colorimetric assay. The color intensity is in direct proportion to the iron concentration and was measured spectrophotometrically at 700/570 nm. A particle-enhanced immunoturbidimetric assay was used for the quantification of ferritin. The precipitate was determined using a turbidimetric method at 570/800 nm.

The results of the analyses of the biochemical parameters and the test results were expressed as mean values ± standard deviation. The normal distribution of the data for ADHDT, WISC-R test, Hcy, vitamin B12, vitamin D, iron, and ferritin values was checked using the Kolmogorov–Smirnov test. The differences between the mean values were tested using the Student’s t-test. A two-way analysis of variance (ANOVA) was performed to statistically analyze the measured biochemical parameters and test results. The following factors were considered as possible sources of variability in the monitored parameters: (1) group (HC and ADHD); (2) age (four groups: 6.5–8.0, 8.08–9.6, 9.7–11.2, 11.3–12.75 years); and (3) the interaction of group and age. Pearson correlation was used to evaluate the association between the ADHDT and WISC-R test scores and biochemical parameters in the children with ADHD. A multiple logistic regression model was used to examine the functional relationship between biochemical parameters and ADHD diagnosis. This model was used to identify which specific biochemical parameters could potentially indicate the presence of ADHD symptoms. The log odds ratio derived from this analysis served as a quantifiable measure of the likelihood of male participants exhibiting ADHD symptoms based on the parameter measured. Logistic regression was used to identify significant predictors of ADHD diagnosis among the measured biochemical parameters, providing a more nuanced understanding of the diagnostic landscape. All statistical analyses were performed using SPSS statistical software (version 20.0.0; IBM, Armonk, NY, USA) and Statistical advanced analytics software (version 8.0; StatSoft, Tulsa, OK, USA).

## 3. Results

First, we investigated whether there were age discrepancies between the ADHD and HC groups to ensure that age did not influence the observed results. The 66 subjects in the HC group had a mean age of 9.94 ± 1.52 years, while the 67 subjects with ADHD had a mean age of 10.13 ± 1.41 years. The subjects in the HC group and those with ADHD did not differ significantly in age (t = −0.756, df = 131, *p* > 0.05). The mean scores for the ADHDT and all WISC-R tests are shown in Table 1.

The two-way ANOVA showed a remarkable difference in the results between the ADHD and HC groups (Table 2). Individuals diagnosed with ADHD had higher scores on the ADHDT, while they scored lower on all WISC-R assessments compared to the HC cohort. Remarkably, there were no significant differences in the results of both tests with respect to age.

Significant differences between the ADHD and HC groups were observed in the mean values of all measured biochemical parameters, as determined by two-way ANOVAs (Table 3). Of note, age did not contribute significantly to these differences. It is noteworthy that the mean values of serum levels of biochemical parameters in each group were within typical values for children, indicating consistency with expected physiological norms.

The serum levels of Hcy, iron, and ferritin were significantly higher in the ADHD patients (respective statistical values of: F_0.001;1;125_ = 46.27; F_0.001;1;125_ = 11.307; F_0.05;1;125_ = 5.95), while vitamin B12 levels were significantly lower compared to the HC group (F_0.01;1;125_ = 9.83), as revealed by the two-way ANOVAs. Vitamin D levels were not significantly different between the HC and ADHD groups (F_0.05;1;125_ = 0.123). Both groups had significantly lower vitamin D levels compared to the reference value of 80 nmol/L (t = −7.928, df = 131, *p* < 0.001 for HC, and t = −5.714, df = 132, *p* < 0.001 for ADHD). Further analyses using Pearson correlation coefficients (Table 4) revealed a significant negative correlation between the serum levels of vitamin D (*p* < 0.05) and ADHDT scores.

The results of the multiple logistic regression model showed that Hcy is associated with the severity of ADHD symptoms (Table 5). Compared to the HC group, the ADHD group had a significantly higher odds ratio for Hcy (B = −1.245, standard error (SE) = 0.251, *p* < 0.001). In addition, the coefficient for iron was significantly greater (B = 0.145, SE = 0.056, *p* < 0.01). The odds ratio of 1.156 indicates that every 1 unit more of iron is associated with a decrease in ADHD symptoms (Table 5). In the ADHD group, compared to the HC group, both vitamin B12 and vitamin D had lower odds ratios (0.997 and 0.996, respectively). The negative coefficient values observed for both vitamin B12 and vitamin D imply that elevated levels of these vitamins are associated with a lower likelihood of an ADHD diagnosis.

## 4. Discussion

This study provides new insights into the serum concentrations of Hcy, vitamin B12, vitamin D, ferritin, and iron in a cohort of male subjects aged 6.5–12.5 years with ADHD in Serbia. A special aspect of this study is that it is the first research in Serbia that has found a remarkable increase in Hcy levels with a simultaneous significant decrease in vitamin B12 levels in children with ADHD. In addition, our analysis shows a significant inverse association between decreased vitamin D levels and increased ADHDT scores, shedding new light on the interplay between this essential vitamin and ADHD symptomatology in this population. These findings emphasize the unique contribution of this study to a better understanding of the biochemical profile associated with ADHD in the Serbian context.

Elevated Hcy concentrations are widely recognized for their causative role in neurological damage due to their neurotoxic effects [54,55]. Elevated Hcy levels are thought to trigger neuronal damage, thereby altering the structural integrity of the brain [28,47,56]. In addition, elevated Hcy levels lead to excitotoxicity and increase oxidative stress, which, in turn, contributes to the occurrence of psychiatric disorders [47,56,57]. Hyperhomocysteinemia triggers the production of neurotoxic by-products, namely homocysteic acid and cysteine sulphonic acid, which damage neurons [58]. Bailey et al. [59] described Hcy levels in healthy children and adolescents and established reference intervals for subjects aged 7–12 years of 3.4–8.4 μmol/L. However, it is important to recognize that Hcy levels can be influenced by various factors, including age, gender, ethnic differences, vitamin B12 deficiency, folate metabolism, and the methods used to determine Hcy in children and adolescents [59,60,61]. The study conducted by Altun et al. [62] showed a reduced Hcy level in patients diagnosed with ADHD in contrast to healthy controls. Conversely, divergent results from other studies emphasize the importance of elevated Hcy levels in the pathogenesis of ADHD [63,64,65]. In line with this discourse, our study revealed a statistically significant increase in Hcy levels in individuals diagnosed with ADHD compared to the HC cohort. Our observation that Hcy levels were significantly higher in the ADHD group than in the HC cohort suggests a plausible association between elevated Hcy levels and the manifestation of ADHD. Importantly, to our knowledge, there are no comparable studies in our region or in neighboring countries that refer to pediatric ADHD populations, which limits the possibilities for a direct comparative analysis.

Vitamin B12 is essential for various neurological developmental processes, including brain maturation, nerve myelination, and cognitive abilities [32,66]. Its central role also extends to the prevention of disorders associated with central nervous system development, mood disorders, and cognitive decline [67,68]. Vitamin B12 deficiency may contribute in a complex manner to the etiopathogenesis and clinical manifestations of emerging neurological disorders [69]. Oh and Brown [70] point out that vitamin B12 is instrumental in the conversion of Hcy to methionine, a process that is critical to metabolic integrity. In particular, a deficiency of vitamin B12 or folate can lead to elevated Hcy levels, a phenomenon confirmed by Pavlovic’s study [67]. Normally, reference values for vitamin B12 in healthy cohorts are below 350 pmol/L [71]. Notable observations indicate decreased vitamin B12 levels in individuals with ADHD compared to healthy controls [72]. Our results mirror this trend and show significantly lower vitamin B12 levels in children with ADHD compared to their HC counterparts. This is in line with previous research by Yektas, Alpay, and Tufan [32], which emphasizes the prevalence of increased Hcy levels and decreased vitamin B12 concentrations in children with ADHD. Similarly, Saha et al. [64] suggest an association between hyperactive and impulsive behavior in ADHD and vitamin B12 deficiency and hyperhomocysteinemia. A growing number of publications [22,24,26] emphasize the links between elevated Hcy levels, reduced levels of vitamins B12 and B6, and the development or symptoms of ADHD. Therefore, the importance of vitamin supplementation in the treatment of neurodevelopmental disorders, including ADHD, should be further investigated. Vitamin B12 is involved in the synthesis of neurotransmitters such as serotonin and dopamine, which play a key role in regulating mood, attention, and impulse control—functions that are often impaired in people with ADHD [73]. One possible mechanism by which vitamin B12 supplementation may help in the treatment of ADHD, therefore, is its role in regulating Hcy levels. Elevated Hcy levels are associated with neurotoxicity, oxidative stress, and neuronal damage, all of which may contribute to the pathophysiology of ADHD. Vitamin B12 supplementation could, therefore, help to mitigate the neurotoxic effects of elevated Hcy levels in people with ADHD. In addition, vitamin B12 deficiency is associated with cognitive impairment, including memory deficits and a decreased attention span—symptoms that characterize ADHD [74]. By treating vitamin B12 deficiency with supplements, it may be possible to improve cognitive function and attention control in people with ADHD. In addition, vitamin B12 plays a role in DNA synthesis and methylation, processes that are essential for proper brain development and function [75]. Therefore, ensuring adequate vitamin B12 levels through supplementation can promote optimal brain health and reduce the risk of neurodevelopmental disorders such as ADHD. To summarize, vitamin B12 supplementation as part of a comprehensive treatment approach for ADHD could offer several benefits, including better symptom management, improved cognitive function, and better overall outcomes, but the overall efficacy of vitamin B12 supplementation should be investigated in more depth. It is also important to note that the response to vitamin supplementation may vary between individuals and that further research is needed to determine the optimal dosage and duration of supplementation for people with ADHD. In addition, vitamin B12 supplementation should be used in conjunction with other evidence-based treatments, such as behavioral therapy and medication, to achieve the best results for people with ADHD.

Iron serves as a crucial cofactor for enzymes that are important in both the synthesis and degradation of monoaminergic neurotransmitters, suggesting a possible indirect involvement in the pathophysiology of ADHD [76,77]. Iron deficiency is known to trigger abnormal dopaminergic neurotransmission [78] and is thought to contribute to the physiopathology of attention deficit hyperactivity disorder [79]. Intracellularly, iron is safely stored through its binding to ferritin, thereby attenuating the deleterious effects of free iron [80]. Normally, serum iron and serum ferritin levels are measured in a range of 50–120 μg/dL and 7–140 ng/mL, respectively [81]. However, studies investigating iron and ferritin levels in ADHD patients yielded contradictory results [82]. While some studies found no significant difference in ferritin levels between ADHD patients and healthy controls [83,84,85], others reported significantly lower ferritin levels in ADHD patients compared to their unaffected counterparts [79,80,86,87]. In contrast, Wang et al. [88] observed elevated serum ferritin levels in ADHD patients with concomitant psychiatric comorbidities compared to patients without such comorbidities. In the present study, iron and ferritin levels were significantly higher in individuals with ADHD compared to healthy controls. In addition, a study by Degremont et al. [89] suggests that cerebral iron levels, rather than systemic iron levels, may be more relevant to the pathophysiology of ADHD in children. Nevertheless, a broader analysis in different populations is essential to clarify the impact of these parameters on the etiology and symptomatology of ADHD.

In addition to many important functions of vitamin D, such as calcium and phosphorus absorption, the inhibition of parathyroid hormone secretion, and the regulation of bone function [90], recent studies have found that vitamin D is also important for brain development [91]. Therefore, its deficiency can lead to neurodevelopmental disorders such as ADHD [92,93]. The blood level for vitamin D has been defined in most studies as the following: blood levels of less than 50 nmol/L represent vitamin D deficiency and blood levels below 80 nmol/L indicate insufficiency [94,95]. The situation is similar in children; levels below 75 nmol/L indicate vitamin D insufficiency [96]. The results of studies on vitamin D levels in children with ADHD are inconsistent [39,97,98]. A study by Goksugur et al. [99] and Şahin et al. [100] found significantly lower vitamin D levels in children and adolescents with ADHD compared to healthy controls. With regard to vitamin D, our results show that there was no significant difference between the ADHD and HC groups. Considering the reference values, significantly lower vitamin D levels were found in both groups in our study. Factors such as skin color and demographic climate may influence vitamin D levels [101]. If we take this into account, it could explain the significantly lower vitamin D levels in both groups in our study when considering the reference values. In our study, a significant negative correlation was found between vitamin D levels and ADHDT scores. This shows that ADHD children with lower vitamin D levels have higher levels of hyperactive behavior and inattention. In addition, our study showed that with higher WISC-R scores, ADHD inattention and hyperactivity symptoms are lower.

## 5. Conclusions

The results of this study emphasize the potential utility of Hcy as a sensitive, albeit non-specific, indicator for predicting the likelihood of ADHD in children. Individuals with ADHD have significantly elevated levels of Hcy, iron, and ferritin, while vitamin B12 concentrations are significantly lower compared to children with normal development. This study clarifies the significance and specificity of these biochemical markers. Moreover, its significance is enhanced by its pioneering character, as it represents the first study of ADHD in Serbian primary school children. It emphasizes the need for standardized and objective diagnostic protocols, including biochemical indices, in the clinical setting. The early identification of such markers is promising for the implementation of preventive interventions that can significantly alleviate or prevent ADHD symptoms. In addition, the use of dietary supplements alongside pharmacotherapy has the potential to not only alleviate symptoms, but also mitigate adverse effects. Future studies should analyze further biochemical parameters in the blood in order to improve the diagnostic precision of ADHD.

## Figures and Tables

**Table 1 children-11-00497-t001:** ADHDT, WISC-R test scores (mean ± standard deviation) for HC and ADHD groups.

Parameter	HC	ADHD
ADHDT	63.15 ± 27.26	98.81 ± 16.05
WISC-R scores		
– Verbal IQ	110.38 ± 12.84	102.79 ± 12.66
– Non-verbal IQ	112.65 ± 12.64	105.91 ± 12.90
– Total IQ	111.56 ± 12.05	104.04 ± 11.47

ADHDT = Attention Deficit Hyperactivity Disorder Test; WISC-R = Wechsler Intelligence Scale for Children-Revised; HC = healthy control; ADHD = attention deficit hyperactivity disorder; IQ = intelligence quotient.

**Table 2 children-11-00497-t002:** Two-way ANOVA results for ADHDT, WISC-R test scores in respect to group and age.

Parameter	ADHDT				WISC-R Test								
					Verbal IQ			Non-Verbal IQ			Total IQ		
	df	MS	F	*p*	MS	F	*p*	MS	F	*p*	MS	F	*p*
Group	1	33,835	66.491	3.153 × 10^−13^ ***	1478	9.203	0.003 ***	1861	12.103	0.001 ***	1815	15.039	1.692 × 10^−4^ ***
Age	3	337.3	0.663	0.576	296	1.845	0.142	222	1.443	0.233	279	2.312	0.079
Group × age	3	201	0.395	0.757	30	0.19	0.903	291	1.894	0.134	110	0.915	0.436
Error	125	508.9			161			154			121		

*** *p* < 0.001; ANOVA = analysis of variance; df = Degrees of freedom; MS = Mean squares; F = variation between sample means/variation within the samples; ADHDT = Attention Deficit Hyperactivity Disorder Test; WISC-R = Wechsler Intelligence Scale for Children-Revised; IQ = intelligence quotient.

**Table 3 children-11-00497-t003:** Serum levels (mean ± standard deviation) of biochemical parameters for HC and ADHD groups.

Parameter	HC	ADHD
Iron (μmol/L)	13.75 ± 3.78	16.99 ± 5.69
Ferritin (ng/mL)	32.72 ± 15.51	40.39 ± 18.67
Hcy (μmol/L)	6.58 ± 1.15	8.78 ± 1.75
Vitamin B12 (pg/L)	469.87 ± 174.37	359.66 ± 174.58
Vitamin D (nmol/L)	64.97 ± 15.52	64.16 ± 22.59

HC = healthy control; ADHD = attention deficit hyperactivity disorder.

**Table 4 children-11-00497-t004:** Pearson correlations of biochemical parameters with age and ADHDT, WISC-R test scores for ADHD.

Parameter	Age		ADHDT		WISC-R Test					
					IQ Verbal		IQ Non-Verbal		IQ Total	
	r	*p*	r	*p*	r	*p*	r	*p*	r	*p*
Iron	0.050	0.689	−0.102	0.410	0.086	0.487	0.039	0.753	0.105	0.400
Ferritin	−0.085	0.496	−0.021	0.863	0.055	0.660	−0.164	0.185	−0.062	0.620
Hcy	0.211	0.086	−0.233	0.058	−0.048	0.703	0.061	0.626	−0.006	0.960
Vitamin B12	−0.135	0.276	0.186	0.132	−0.039	0.756	−0.029	0.819	−0.057	0.647
Vitamin D	−0.093	0.456	−0.267	0.029 *	−0.098	0.430	−0.207	0.092	−0.216	0.079

* *p* < 0.05; ADHDT = Attention Deficit Hyperactivity Disorder Test; WISC-R = Wechsler Intelligence Scale for Children-Revised; ADHD = attention deficit hyperactivity disorder; IQ = intelligence quotient.

**Table 5 children-11-00497-t005:** Multiple logistic regression model results for ADHD group compared to HC group.

Parameter	B	SE	Wald χ^2^ Test	df	*p*	Exp(B)
Iron	0.145	0.056	6.792	1	0.009 **	1.156
Ferritin	0.028	0.015	3.393	1	0.065	1.029
Hcy	1.245	0.251	24.683	1	0.000 ***	3.474
Vitamin B12	−0.003	0.002	2.879	1	0.090	0.997
Vitamin D	−0.004	0.013	0.082	1	0.775	0.996

** *p* < 0.01; *** *p* < 0.001; ADHD = attention deficit hyperactivity disorder; SE = standard error; df = degrees of freedom; Exp(B) = exponentiation of the B coefficient.

## Data Availability

The raw data supporting the conclusions of this article will be made available by the authors on request.

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
