# Peer review of "Serum Biomarker Analysis in Pediatric ADHD: Implications of Homocysteine, Vitamin B12, Vitamin D, Ferritin, and Iron Levels"

_children, 2024, doi:10.3390/children11040497_

Round 1
Reviewer 1 Report
Comments and Suggestions for Authors
Page 6 Vitamin D levels were not significantly different between HC and ADHD (F0.05, 1, 125 = 0.123). Those diagnosed with ADHD had slightly lower 228 vitamin D levels compared to the HC.
If not significantly lower then NOT lower – can’t refer to differences that do not reach significance as differences:
Vitamin D levels were not significantly different between HC and ADHD. PERIOD. FULL STOP
Page 6:
The serum levels of Hcy, iron and ferritin were significantly higher in the ADHD 224 patients (respective statistical values of: F0.001, 1, 125 = 46.27; F0.001 ,1,125 = 11.307; F0.05,1,125 = 5.95), 225 while vitamin B12 levels were significantly lower compared to HC (F0.01, 1, 125 = 9.83), as
Page 8:
In the present study, iron and ferritin levels tended to be higher in individuals with ADHD compared to healthy controls, 317 although not statistically significant.
Which is it with regard to iron and ferriting Yes significantly higher or not and ,AGAIN, if not significantly different then NOT DIFFERENT:
From a statistical point of few, does there have to be a correction for multiple comparisons?
From table 3:
|
Hcy μmol/L |
6._5_8_ _± _1_._1_5_ _ |
8_._7_8_ _± _1_._7_5_ _ |
In text page 7
Our observation that Hcy levels were 3.468 times higher in the 279 ADHD group than in the HC cohort suggests
8.78 is NOT 3.468 TIMES higher than 6.58 it is 1.33 times higher or 33% higher.
Where does that 3.468 TIMES number come from?
Author Response
Dear Reviewer,
Many thanks on your comments that will improve our manuscript.
- Page 6 Vitamin D levels were not significantly different between HC and ADHD (F0.05, 1, 125 = 0.123). Those diagnosed with ADHD had slightly lower 228 vitamin D levels compared to the HC. If not significantly lower then NOT lower – can’t refer to differences that do not reach significance as differences: Vitamin D levels were not significantly different between HC and ADHD. PERIOD. FULL STOP
We accept your suggestion and have corrected the manuscript accordingly.
- Page 6: The serum levels of Hcy, iron and ferritin were significantly higher in the ADHD patients (respective statistical values of: F0.001, 1, 125 = 46.27; F0.001 ,1,125 = 11.307; F0.05,1,125 = 5.95), while vitamin B12 levels were significantly lower compared to HC (F0.01, 1, 125 = 9.83), as Page 8: In the present study, iron and ferritin levels tended to be higher in individuals with ADHD compared to healthy controls, although not statistically significant. Which is it with regard to iron and ferriting Yes significantly higher or not and ,AGAIN, if not significantly different then NOT DIFFERENT
We apologise for the omission. We have checked the results and corrected the statement on page 8 in the Discussion section.
- From a statistical point of few, does there have to be a correction for multiple comparisons?
Corrections for multiple comparisons are not necessary because each comparison asks the same question: were the values of the biochemical parameters lower or higher in the ADHD group? The conclusion for all comparisons was that the ADHD group had lower or higher values for some biochemical parameters compared to the healthy group.
4. From table 3: Hcy μmol/L was 6._5_8_ _± _1_._1_5_for HC and 8_._7_8_ _± _1_._7_5_ for ADHD group In text page 7 Our observation that Hcy levels were 3.468 times higher in the ADHD group than in the HC cohort suggests 8.78 is NOT 3.468 TIMES higher than 6.58 it is 1.33 times higher or 33% higher.
Corrections were made in this sentence.
Where does that 3.468 TIMES number come from?
We have made a typing error. The correct number is 3.474, as Table 5 shows. This logarithmic ratio (Exp(B) in Table 5) represents the value of the probability that male participants have ADHD symptoms for the parameter measured.

Reviewer 2 Report
Comments and Suggestions for Authors
The study by Lukovac et al. explores the correlation between homocysteine levels and ADHD in primary school-aged boys, presenting a comprehensive analysis of serum biomarkers such as homocysteine, vitamin B12, vitamin D, ferritin, and iron. The study's robust methodology includes a well-defined cohort of 133 subjects, differentiated into those diagnosed with ADHD and healthy controls, thereby ensuring the reliability of its findings. One of the significant contributions of this research is its identification of significantly increased levels of homocysteine and decreased levels of vitamin B12 in ADHD patients compared to controls, highlighting a potential biomarker for ADHD diagnosis and prognosis.
A minor suggestion for revision involves expanding on the implications of these findings in the context of nutritional interventions and the potential for incorporating vitamin B12 supplementation as a complementary approach to managing ADHD symptoms. Additionally, further clarification on the statistical methods, particularly the criteria for choosing the multiple logistic regression model parameters, could enhance the readers' understanding of the analytical framework. This study marks a significant step forward in ADHD research, particularly in the Serbian context, and sets the stage for future investigations into nutritional and biochemical interventions for ADHD management.
Author Response
Dear Reviewer,
Thank you for your valuable comments that significantly improved our manuscript.
- A minor suggestion for revision involves expanding on the implications of these findings in the context of nutritional interventions and the potential for incorporating vitamin B12 supplementation as a complementary approach to managing ADHD symptoms.
Thank you for your comment. We have expanded the discussion as you suggested (line 324).
- Additionally, further clarification on the statistical methods, particularly the criteria for choosing logistic the multiple regression model parameters, could enhance the readers' understanding of the analytical framework.
Thank you for this advice. In the section „Materials and methods“, line 196, we have improved the part about the multiple logistic regression used.

Reviewer 3 Report
Comments and Suggestions for Authors
Dear authors:
I congratulate you on the relevance of your research. Your article is rigorous in its methods and in its analysis of results.
My comments to improve it:
-It seems to me that the database search could have been more exhaustive. Only 7 references are from the last 4 years. In a brief search of PUBMED, I found several publications from the last 3 years.
-Another note I'd like to make is about the abstract. The abstract must have an opening sentence that sets out framing with the title (ADHD vs increased homocysteine levels) and one sentence that clearly describes: purpose of research; characteristics of research and type of research.
-About the title: In my opinion, it's not entirely in line with the research you've done. It is only a general problem-issue. I'd ask you to rethink that.
-Revisit the section “2. Materials and Methods”: provide a comprehensive overview of the research methodology (type of research) and design research you followed. This theoretical conceptual framework should begin the section.
Rew
Author Response
Dear Reviewer,
Thank you for your comments that helped us improve our manuscript.
- It seems to me that the database search could have been more exhaustive. Only 7 references are from the last 4 years. In a brief search of PUBMED, I found several publications from the last 3 years.
We conducted a more extensive search for published research articles and reviews and added 15 additional references to the manuscript, all published in the last 3 years.
- Another note I'd like to make is about the abstract. The abstract must have an opening sentence that sets out framing with the title (ADHD vs increased homocysteine levels) and one sentence that clearly describes: purpose of research; characteristics of research and type of research.
The abstract has been modified according to the suggestions in this comment. We have added an introductory sentence to the abstract that emphasises the importance of the data presented. We have also clearly outlined the aims and characteristics of this research.
- About the title: In my opinion, it's not entirely in line with the research you've done. It is only a general problem-issue. I'd ask you to rethink that.
We agree with your opinion and have changed the title according to our research.
- Revisit the section “2. Materials and Methods”: provide a comprehensive overview of the research methodology (type of research) and design research you followed. This theoretical conceptual framework should begin the section.
Thank you for your suggestion. We have revised the section and expanded some parts to justify the methodological choices we made for this study. We have also added an introductory paragraph, as you suggested, in which we define the nature of the research and the experimental design (a correlational study with a cohort design).

Round 2
Reviewer 1 Report
Comments and Suggestions for Authors
The authors have addressed the issues I raised in my initial review. There are 2 sentences that I think should be less categorical.:
Page 6 line 251-252 “important indicative parameter”. This is still an association so need to describe it as such.
Similarly on page 8 313-314 recommending supplementation with B12 is not a clinically proven intervention and to describe it at as something of “importance that cannot be overestimated” is an Overstatement. This is not proven therapy. Probably not harmful and it can be proposed as something warranting investigation (and I may try it) but this phrasing makes it sound like standard of care that must be done and most are not doing it.
There’s a saying among entrepreneurs that it is important not to get oversold on your own idea.
Author Response
Dear Reviewer,
Thank you for your comments.
On line 251, we have changed “important indicative parameter” to "is related to the severity of ADHD symptoms" as you suggested.
On page 8 of the Discussion, we have made some changes to make it clearer that vitamin B12 supplementation in ADHD treatment needs further investigation and that it is not standard clinical practise (lines 315, 325, 331, 333, in green).

Reviewer 3 Report
Comments and Suggestions for Authors
Dear Authors:
I very much appreciated the improvements to the article.
I have nothing further to comment.
Rew
Author Response
Dear Reviewer,
Many thanks!
Kind regards
